# Microbiology and Antimicrobial Resistance Profile in Patients with Diabetic Foot Sepsis at a Central Hospital in Johannesburg, South Africa

**DOI:** 10.3390/diagnostics15010032

**Published:** 2024-12-26

**Authors:** Simran Patel, Emeline Jooste, Charalambia Glynos, Onyiyechukwu Mbajiorgu, Anelisa Sipahlanga, Wandile Ngubane, Gopala Maharaj, Maeyane Stephens Moeng, Thifhelimbilu Emmanuel Luvhengo

**Affiliations:** 1Unit of Undergraduate Medical Education, University of the Witwatersrand, Johannesburg 2017, South Africa; simy.p12@gmail.com (S.P.); emelinejooste@gmail.com (E.J.); charaglynos@gmail.com (C.G.); mbajior@gmail.com (O.M.); lisasipahlanga@gmail.com (A.S.); wandilengubane311@gmail.com (W.N.); gopala.maharaj1@students.wits.ac.za (G.M.); 2Department of Surgery, Charlotte Maxeke Johannesburg Academic Hospital, Johannesburg 2193, South Africa; maeyane.moeng@wits.ac.za

**Keywords:** diabetic foot sepsis, microbiology, antibiotics, resistance, South Africa

## Abstract

**Background**: Diabetic foot sepsis (DFS) is the leading cause of lower extremity amputations and timely initiation of effective antimicrobial therapy is paramount during its management. This study investigated causative microorganisms and their antimicrobial susceptibility profile in patients with DFS. **Materials and Methods**: A retrospective review was conducted on patients who were 18-years and older admitted with DFS. Data collected included demographic information, comorbidities, clinical findings, types of specimens collected and results of microscopy, culture, and sensitivity (MC&S), treatment, and outcomes. **Results**: One hundred and sixty-eight records were found, of which 64.3% were of male patients. The median (IQR) age of males was 58 years (IQR 54–65) compared to 61 years (IQR 54–67) for females. Results of MC&S were available in 63.1% of the records, and *E. faecalis* was cultured in 16%, *P. mirabilis* in 10%, and *S. aureus* in 8% of cases. Amoxicillin/Clavulanic acid was prescribed in 69% of the cases. Resistance to at least one antimicrobial was shown in 88% of *S. aureus* and 80% of *P. mirabilis species*. **Conclusions**: The commonly cultured organisms in patients with DFS were *E. faecalis* 16%, *P. mirabilis* 10%, and *S. aureus*. Amoxicillin/Clavulanic was prescribed empirically in 69% of the cases despite high rates of resistance, and in 37% treatment was not preceded by collection of specimens for MC&S. We therefore recommend collection of specimens for MC&S before initiation of antimicrobial therapy in all patients with DFS.

## 1. Introduction

Diabetes mellitus (DM) is among the common non-communicable diseases and its incidence is increasing, especially in African countries, including South Africa [1]. Diabetic foot syndrome is defined by the World Health Organization as a severe complication of DM characterized by foot ulceration associated with neuropathy, varying degrees of ischemia, and infection [2]. Foot ulceration develops in around 19–34% of patients with DM and may lead to major amputations and/or secondary infection [3]. Foot infection in patients with DM is referred to as diabetic foot sepsis (DFS), which is more common in male patients with type 2 DM [4]. The prevalence of DFS in Africa exceeds the global average and leads to comparatively higher rates of major amputations and mortality on the continent [5,6,7,8,9].

Most cases of DFS are preceded by a neuropathic, ischaemic, or neuro-ischaemic ulcer. Other factors that contribute to the development of foot ulcerations in patients with DM include poor glycaemic control, host-related elements, and microbial characteristics [4,9]. Several species of bacteria and sometimes fungi are involved in DFS, either in isolation or as synergistic infections. It is crucial that, during the investigation of patients with diabetic foot ulcers, an adequate specimen be collected for MC&S to identify causative organisms and their antibiograms [3,10,11,12]. Empiric antimicrobial therapy should be initiated in a timely manner while awaiting MC&S results to prevent worsening of the already clinically overt infection.

Management of DM and its complications in low- and middle-income countries (LMICs), including South Africa, faces significant challenges, among them poor understanding of the disease; this results in a lot of patients “suffering from” instead of “living with” diabetes. Consequently, majority of patients in LMICs experience inadequate self-care, poor dietary habits, and reduced physical activity, and often manifest suboptimal health-seeking behaviour [13]. Additionally, the microbial profiles and antimicrobial susceptibility patterns of patients with DFS in LMICs are often different to those in high-income countries [14]. Compounding these challenges is the high prevalence comorbidities like HIV, which increases risk of DM and the likelihood of severe soft tissue infections [15]. Limited financial resources in LMICs may also limit access to healthy food, resulting in over-reliance on inexpensive carbohydrate-dense diet, which is diabetogenic [15].

The frequently isolated organisms in patients with DFS are aerobic gram-positive cocci, with *S. aureus* being the most predominant species [16]. Polymicrobial and chronic infections often involve anaerobic organisms and gram-negative aerobic bacilli such as *P. aeruginosa* and *Enterobacteriaceae* [11,17]. Emergence of antimicrobial resistance is a growing concern. Various strains of bacteria, including extended-spectrum β-lactamase-producing *Enterobacteriaceae*, methicillin-resistant *S. aureus *(MRSA), specific *P. aeruginosa* strains, and vancomycin-resistant enterococci, are frequently isolated in specimens collected for MC&S from patients with DFS [11,16,17].

Obtaining high-quality and representative samples for MC&S followed by targeted antibiotic therapy is crucial for appropriate and effective management of patients with any infection, including DFS [12,18]. Collecting specimens for MC&S also provides data on local susceptibility patterns for use as a guide by clinicians during selection of appropriate empirical antimicrobial therapy [19,20]. By integrating antibiogram data into clinical practice, clinicians can reduce the misuse of antibiotics and contribute to the effort to curb an increase in the incidence of antimicrobial resistance.

Several methods are used to obtain samples from patients with DFS among them deep tissue biopsy, sampling adjacent bone near the infection site, pus swabs, blood cultures, and fluid aspiration. While the gold standard is deep tissue biopsy, especially when osteomyelitis is suspected, it is acceptable to collect superficial samples if the infection is limited [21,22,23].

Several classification systems are used to grade severity and to guide management of patients with DFS but none of the classification systems is supreme and universally accepted [24]. The choice of a classification system for grading of DFS depends on a clinician as well as its usability, as some of the classification systems are complex [24]. It is advisable to use a severity grading system that includes the extent of the severity of infection, degree of ischaemia, causative organisms, and the status of glycaemic control [24]. Superficial infection is likely to be caused by one species of bacteria whereas deep and severe DFS is likely to be caused by a combination of aerobic and anaerobic organisms. The severity of DFS is also used to guide the duration of antimicrobial therapy [25].

Ultimately, effective treatment of DFS hinges on a comprehensive understanding of the microbiology and susceptibility patterns of microorganisms commonly associated with DFS. Evaluating antimicrobial utilization and management trends in LMICs is critical for timeous and effective treatment strategies. Resources in the majority of LMICs are limited, patients often present late, and antimicrobial stewardship is not routine. This study aimed to identify causative organisms and their susceptibility profiles in patients presenting with DFS.

## 2. Patients and Methods

This was a retrospective cross-sectional observational study focusing on patients aged 18 years and older who were admitted to the Department of Surgery at Charlotte Maxeke Johannesburg Academic Hospital (CMJAH) with DFS from 1 January 2017 to 31 December 2019. The study included patients with both primary and recurrent DFS. Data were obtained from weekly morbidity and mortality (M&M) meeting reports stored in the Research Electronic Data Capture (REDCap) database, discharge summaries, and hospital admission files. Data collected included age, sex, race, specimen collected for MC&S and results, antimicrobial(s) prescribed, and overall outcome. The outcome data collected included severity of infection, type of surgical intervention, and the level of amputation.

Descriptive statistical analysis was carried out using STATA SE 17.0 software. Categorical data, such as types of specimens collected, antimicrobials prescribed, and resistance patterns of organisms, were presented as actual counts and percentages. The study also examined resistance to antimicrobials in relation to gender and race, with confidence intervals determined using the exact method. Ethical clearance for the study was obtained from the Human Research Ethics Committee of University of the Witwatersrand (M210943). Consent from individual patients was waived as it was a retrospective study. The study was conducted following guidelines contained in the declaration of Helsinki.

## 3. Results

Hundred and sixty-eight (168) records met the inclusion criteria. The median age of the included patients was 59 years (IQR 54–67). The median age of male patients was 58.9 years compared to 61 years for females. Eighty-nine (53%) of the patients were black Africans while 26% (44) were White. The majority, 94.6%, of the patients had type II DM. Most cases, 95.8%, presented with wet gangrene of the toes or mid-foot (Table 1).

The median WCC of male patients was 11 compared to 10.6 for females, but the difference was not statistically significant (*p* = 0.438). The median HbA1c in male and female patients were 9.8% and 11.4%, respectively and the difference was not statistically significant (*p* = 0.266). One hundred and six (63.1%) had records of the type of specimens collected for MC&S and their results. The collected specimens for MCS included tissue samples in 60.4% (64) of the patients (Table 2).

Among the top five commonly isolated organisms were *E. faecalis* (16%), *P. mirabilis* (10.4%), *S. aureus* (7.5%), *P. aeruginosa* (6.6%), and *K. pneumoniae* (4.7%) (Figure 1).

Of the top 11 identified organisms, *K. pneumoniae*, *Morganella morganii*, and *A. baumanii* exhibited the highest proportion of antimicrobial resistance, with 100% resistance. *S. aureus* demonstrated resistance rates of 88%, *P. mirabilis* 80%, and *P. aeruginosa* 71%. *E. faecalis*, the most commonly isolated organism, displayed the least resistance at 6.7%.

Records of antimicrobial prescriptions were complete in 45.2% (73/168) of the reports and amoxicillin/clavulanic acid was prescribed for 69% (50/73) of the patients with DFS (Figure 2).

Ertapenem was prescribed in 11% of cases, followed by vancomycin in 10%, and piperacillin/tazobactam and cefepime in 5% each. Other antibiotics and combination therapies were prescribed in less than 2% of cases each. Ampicillin/amoxicillin was tested against the top 11 most cultured organisms and demonstrated 100% resistance against most of them and 8.3% against *E. faecalis*. All cases of *S. marcescens* and *M. morganii* were resistant to amoxicillin-clavulanic acid while *K. pneumoniae*, *P. mirabilis*, and *E. coli* displayed resistance rates of 80%, 50%, and 25%, respectively. Ertapenem, the second most commonly prescribed antibiotic, exhibited resistance in 50% of *E. coli* and *K. pneumoniae.* All *P. mirabilis* and *M. morganii* species were sensitive to ertapenem. All species of *P. mirabilis*, *P. aeruginosa*, *E. coli*, and *A. baumanii* were resistant to cefepime compared to 75% *K. pneumoniae* and 50% of *S. marcescens.* Resistance to tazobactam-piperacillin was shown in 100% each of *K. pneumoniae* and *A. baumannii*. Similarly, 50% each of *P. aeruginosa* and *M. morganii* species were resistant to tazobactam-piperacillin (50%). All cases of *P. mirabilis* were sensitive to tazobactam-piperacillin.

Gram-negative specimens displayed 100% resistance to ampicillin/amoxicillin, gentamycin, ampicillin, ceftriaxone, cefuroxime, cefoxitin, trimethoprim, and trimethoprim/sulfamethoxazole. They also showed some degree of resistance to other common antimicrobials such as amikacin, tobramycin, amoxicillin, amoxicillin/clavulanic acid, piperacillin/tazobactam, cefepime, cefotaxime, ceftriaxone, cefoxitin, ceftazidime, ciprofloxacin, and tigecycline, with at least one bacterial sample demonstrating 100% resistance to these antimicrobials. Ertapenem and imipenem showed promising results, with resistance levels recorded at 50% and 60%, respectively, for gram-negative specimens. Gram-positive bacteria exhibited lower resistance rates per antibiotic. All *S. aureus* and *E. faecalis* species cultured showed resistant to ampicillin. None of the *S. aureus* cultured showed resistance to macrolides, clindamycin, cefotaxime/ceftriaxone, and trimethoprim/sulfamethoxazole.

Polymicrobial infections were recorded in 79% of male patients who had MC&S results compared to 50% in females. The commonly isolated organisms in males were *E. faecalis* at 6.9%, *S. aureus* at 9.2%, *K. pneumoniae* at 7.7%, and *P. mirabilis* at 7.7%. In females, *E. faecalis*, *P. mirabilis*, and *aeruginosa* were cultured in 14.3% each, followed by *S. agalactiae* at 8.6% and *S. aureus* at 5.7%. Notably, no isolates of *S. marcescens* were found in females (Table 3 and Table 4).

The top three organisms isolated varied based on race. In the Black population, the most common organisms were *E. faecalis* at 18.5%, *P. mirabilis at* 9.3%, and *K. pneumoniae* at 7.4%. Among Caucasians, *S. aureus* at 20% was the most commonly cultured bacteria, followed by *P. mirabilis* at 15%, and then 10% each for *E. faecalis*, *P. aeruginosa*, and *M. morganii*. In the Indian population, the most frequently isolated organisms were *P. aeruginosa* at 18.2%, followed *by E. faecalis* at 13.6%, and 9% each for *P. mirabilis*, *K. pneumoniae*, and *E. coli.*

Outcomes were analysed based on the microbial profile of ulcers, monomicrobial vs. polymicrobial. Among patients who had MC&S results and had no amputations (8), 75% had polymicrobial while 25% had monomicrobial infections. Among those who underwent amputations (40), 65% had polymicrobial and 35% had monomicrobial infections. The difference in the rate of monomicrobial and polymicrobial infections in patients who had amputation was not statistically significant (*p* = 0.524).

## 4. Discussion

The increasing prevalence of DM combined with improved management has led to more people living longer and an elevated risk of complications, including DFS [21]. The mean age of patients with DFS in the current study was 59 years, falling within the range reported in similar studies conducted in Kenya and the United States [24,25]. Notably, a higher proportion of patients were in the 45–64 years age group, aligning with the mean age in this study [21]. The age of male patients was younger than of females. The earlier onset of complications in males may be attributed to multiple risk factors, including concurrent conditions such as hypertension, smoking, and poor glycaemic control, as well as bad healthcare-seeking behaviours [21,26]. However, these factors were not extensively examined in this study.

The prevalence of polymicrobial infections in DFS patients in this study was high, with 77% of cases presenting with infections due to more than one species of bacteria. The high incidence of polymicrobial infections could be due to the high prevalence of severe and deep infections at presentation [26]. Higher rates of polymicrobial infections observed in our study are consistent with findings in other studies that reported on patients that had deep infections and complex infections [26,27,28]. Close to 96% (95.8%) of patients in the study had wet gangrene. Furthermore, the rate of polymicrobial infections in the current study was higher in male compared to female patients. The observed differences in polymicrobial infections between men and women could have been due to variations in exposure to environmental pathogens and differences in health seeking behaviour, as well as cultural factors [11,17]. While external factors like pre-hospital care and hygiene practices might have influenced the diversity of organisms based on race, these were not studied.

The most isolated organism in the study was *E. faecalis*, which differs from the findings from a meta-analysis conducted by McDonald et al., where *S. aureus* was identified as the predominant isolate in diabetic foot infections [29]. This variation may be due to patients in the quaternary hospital setting presenting at more advanced stages of the disease being more severely immunocompromised, which could allow unusual commensals like *E. faecalis* to predominate [20].

In our study, all isolated microbial species demonstrated resistance to at least three antimicrobial agents. Among the top three organisms, *E. faecalis* exhibited the lowest overall resistance to antimicrobial therapies, highlighting its comparatively more favorable susceptibility profile.

Notably, the resistance patterns of *E. faecalis* in this study align with results from a previous study also conducted in South Africa by Shobo et al., which showed resistance to vancomycin and little resistance to penicillin [30]. This study also showed that *Enterococcus* spp. are also isolated in patients that require prolonged antibiotic therapies and can be mirrored to those with severe DFS, who often get more than one infection and require prolonged treatment. The data also indicate that combination therapy is more effective, with the resistance rate to ampicillin/amoxicillin being 8.3%. This is particularly significant given the propensity of *E. faecalis* to develop resistance [31]. No *S. aureus* were resistant to macrolides, ceftriaxone, cefotaxime, and clindamycin. Resistance patterns between *S. aureus* and *E. faecalis* sometimes overlap [32]. However, the sensitivity of *E. faecalis* to macrolides and clindamycin was not specifically tested in the current study.

Gram-negative bacteria predominated over gram-positives in infections that presented to our tertiary hospital, which is in keeping with findings from LMICs in Asia countries with severe DFS [20]. According to global data, diabetic foot sepsis infections are predominantly caused by gram-positive cocci such as *S. aureus*, including MRSA) and *β-haemolytic Streptococcus* spp. [9,10,33]. Chronic and polymicrobial infections often involve anaerobic and gram-negative aerobic bacilli, including *P. aeruginosa* and *Enterobacteriaceae* [33].

Intriguingly, a significant portion of patients in this study did not receive intravenous antibiotic prescriptions. Intravenous antibiotics antimicrobials were only used in patients with deep infections. This prescription trend is in keeping with a general approach to DFS using the WIFi classification and a multidisciplinary team [34,35,36]. Most sepsis cases in the study were localized to the forefoot and midfoot, and the analysis did not include prophylactic antimicrobials. Some DFS cases were chronic wounds with significant biofilm, and thus were managed with local antiseptic measures and dressing, including negative pressure therapy [37].

It is important to highlight that the top three organisms identified belong to distinct bacterial groups: *Enterococcus* spp., *Streptococcus* spp., and *Enterobacteriaceae* spp. Due to their varying resistance and susceptibility profiles, the use of a broad-spectrum antibiotics is recommended as an initial treatment option. Broad-spectrum antibiotics, which are effective against a wide range of gram-positive and gram-negative aerobes and anaerobes, have been shown to be efficacious against all three groups. Examples include carbapenems and combination antibiotics such as piperacillin-tazobactam and amoxicillin-clavulanate. However, while these agents are critical in initial management, their use should be restricted to empiric therapy. Prompt de-escalation to targeted antibiotics, guided by resistance profiles obtained through proper specimen collection, should be strongly advocated [38]. This approach is crucial for preventing the emergence of multidrug-resistant organisms, as it ensures that more effective antibiotics can be utilized for improved source control [38].

In our research, whether the microbial profile of DFS was monomicrobial or polymicrobial did not have an impact on the likelihood of amputation. This suggests that other factors might have been more influential in determining clinical outcomes in patients with diabetic foot ulcers and may be beyond the scope of this research or better extrapolated from a larger sample size, which would give a better idea of organism susceptibility and resistance patterns that could frame the outcomes better.

This retrospective study presents critical limitations that must be carefully weighed when interpreting the findings. Foremost, the quality and completeness of the data drawn from medical records, discharge summaries, and hospital files may have introduced variations and influenced result robustness. A significant limitation arises from the relatively modest sample size. This limitation raises concerns about the generalizability of the findings, potentially omitting a more comprehensive spectrum of diabetic foot sepsis cases. Furthermore, the non-standardized culture methods employed in the study introduce variability in specimen collection and analysis, potentially impacting result accuracy. This methodological variability may have led to the collection of superficial skin colonizers rather than the specific pathogens associated with diabetic foot sepsis. These limitations underscore the need for future research with more extensive and diverse patient populations, standardized culture methods, and prospective data collection to provide a more comprehensive understanding of diabetic foot sepsis.

## 5. Conclusions

This study provides valuable insights into the management of diabetic foot sepsis (DFS), highlighting several key findings. A notable gender difference was observed, with DFS being more common in males. Amoxicillin/clavulanic acid was the most frequently prescribed antimicrobial, despite over 47% of patients receiving antibiotics without prior specimen collection for MC&S. This underscores the need for more targeted and evidence-based treatment strategies to mitigate the risk of antimicrobial resistance. The microbiological analysis revealed *E. faecalis* and *P. mirabilis* as the most commonly isolated organisms, with more than 67% of the isolates exhibiting resistance to at least one antimicrobial agent. These findings stress the importance of routine specimen collection for MC&S to guide appropriate antibiotic therapy. Moreover, the prevalence of polymicrobial infections and high rates of resistance in this study highlight the need for improved stewardship in antimicrobial use, emphasizing de-escalation strategies after empiric therapy. There was no difference in the rate of amputation in patients with monomicrobial and polymicrobial infection. To better inform clinical practice, future prospective, multicenter studies, particularly in low- to middle-income settings, are essential. Such research can help develop region-specific guidelines and enhance the overall management of DFS, improving patient outcomes while minimizing the threat of multidrug-resistant organisms

## Figures and Tables

**Figure 1 diagnostics-15-00032-f001:**
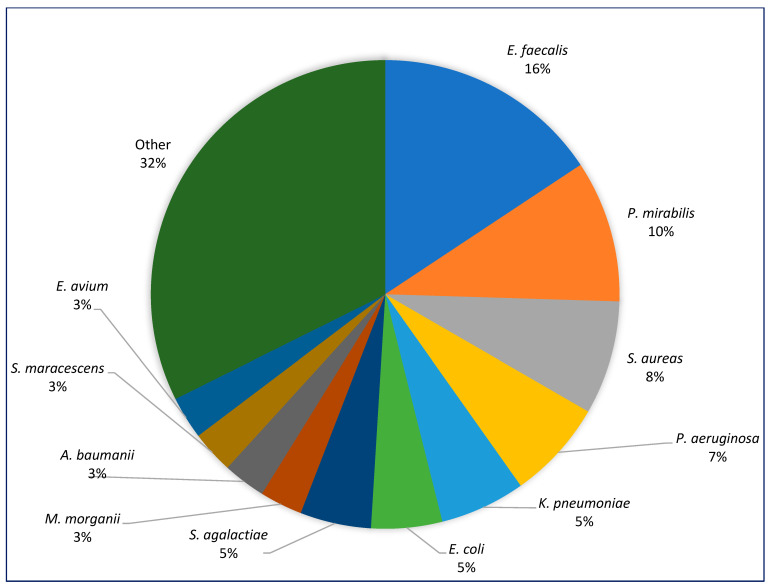
Breakdown of organisms isolated from specimen collected for MC&S in patients with DFS.

**Figure 2 diagnostics-15-00032-f002:**
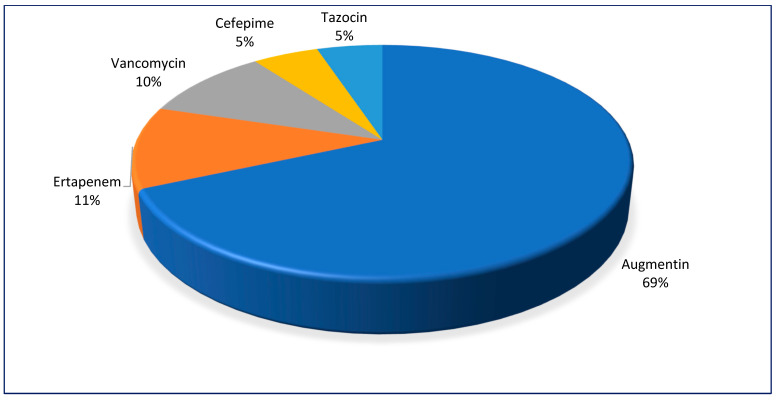
Breakdown of the top five antibiotics prescribed for patients with DFS.

**Table 1 diagnostics-15-00032-t001:** Demography and clinical findings at presentation in patients with DFS (*n* = 168).

Variable	Overall	Male	Female	*p*-Value
Age years (median, IQR)	59 (54–67)	58 (54–65)	61 (54–67)	0.139
Racial group				
African	89 (53%)	54 (50%)	35 (58%)	
White	44 (26%)	32 (30%)	12 (20%)	
Indian	30 (18%)	19 (18%)	11 (19%)	
Coloured	5 (3%)	3 (2%)	2 (3%)	
Type of DM				
Type 1 DM	9 (4.4%)	6 (5.6%)	3 (5%)	0.98
Type II DM	159 (94.6%)	102 (94.4%)	57 (95%)	
Nature of infection				
Wet gangrene/sepsis	161 (95.8%)	101 (93.5%)	60 (100%)	0.398
Ulcers/cellulitis	2 (1.2%)	2 (1.9%)	0 (0%)	
Not specified	5 (3%)	5 (4.6%)	0 (0%)	
Anatomical extent				
Toes	59 (35.7%)	34 (31.5%)	26 (43.3%)	0.67
Mid-foot	94 (56%)	66 (61.1%)	28 (46.8%)	
Ankle	4 (2.3%)	3 (2.8%)	1 (1.6%)	
Below the knee	10 (6%)	5 (4.6%)	5 (8.3%)	

**Table 2 diagnostics-15-00032-t002:** Comparison of laboratory results and types of specimen collected for MCS between male and female patients.

Variable	Overall	Male	Female	*p*-Value
White cell count (×10^9^/L)	10.9 (8.32–15.45)	11 (8.5–14.95)	10.6 (8.2–15.69)	0.438
C-reactive protein (mg/L)	133.5 (5.6.5–247).	153 (57–268)	94 (57–184.5)	0.091
Urea (mmol/L)	7.1 (5.3–9.9)	7.7 (6.0–11.3)	6.7 (3.9–8.53)	0.036
Creatinine (umol/L)	88 (68–136)	100 (68–153.8)	81 (62–102)	0.162
Albumin (g/L)	36 (24–39)	35 (23–40)	37 (30.5–39)	0.109
Haemoglobin A1c (%)	11.1 (8.23–12.6)	9.8 (8.1–12.2)	11.4 (8.85–13)	0.266
Specimen sampling (*n* = 106)				
Tissue	64 (60.4%)	49 (73.1%)	15 (38.5%)	0.018
Pus swab (deep)	15 (14.2%)	7 (10.4%)	8 (20.5%)	
Aspiration	7 (6.6%)	1 (1.5%)	6 (15.4%)	
Blood cultures	20 (18.8%)	10 (14.9%)	10 (25.6%)	
Polymicrobial	77%	79%	50%	
Documented antibiotic treatment	76 (45.2%)	52 (48.1%)	24 (40%)	0.69

**Table 3 diagnostics-15-00032-t003:** List of the organisms cultured in male patients with DFS.

Organism	Percentage
*E. faecalis*	16.9%
*S. aureus*	9.2%
*K. pneumoniae*	7.7%
*P. mirabilis*	7.7%
*M. morganii*	6.2%
*E. coli*	6.2%
*S. marcescens*	2.8%

**Table 4 diagnostics-15-00032-t004:** List of the organisms cultured in female patients with DFS.

Organism	Percentage
*E. faecalis*	14.3%
*P. mirabilis*	14.3%
*P. aeruginosa*	14.3%
*S. agalactiae*	8.6%

## Data Availability

Data supporting results in this study will be made available on request.

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
