# Peer review of "Microbiology and Antimicrobial Resistance Profile in Patients with Diabetic Foot Sepsis at a Central Hospital in Johannesburg, South Africa"

_diagnostics, 2024, doi:10.3390/diagnostics15010032_

Round 1
Reviewer 1 Report
Comments and Suggestions for Authors
Thank you for the opportunity to review this article on microbiological testing and antimicrobial resistance profiles in patients with diabetic foot sepsis at the Central Hospital in Johannesburg, South Africa. The paper appears to be well-researched and thoroughly reviewed. However, I would like to offer the following observations and suggestions for improvement:
- The paper demonstrates that microbiological testing and antimicrobial resistance profiling in patients with diabetic foot sepsis are similar across many developed and developing countries. While this finding is valuable, it seems that little attention has been given to analyzing the differences between these settings. A more in-depth comparison of these differences could provide additional insights and enhance the paper's contribution to the field.
- There appears to be a lack of clarity regarding the nature of bacterial resistance to antibiotics. Specifically, it is not clearly stated whether the bacteria identified in the study are resistant to a single antibiotic or exhibit multi-drug resistance. This distinction is crucial for understanding the complexity of antimicrobial resistance in this patient population. I recommend addressing this aspect more thoroughly in the paper.
- The paper would benefit from a more detailed analysis of the bacterial species most commonly associated with severe outcomes, such as foot amputation or patient death. It would be particularly valuable to include a comprehensive profile of these bacteria and their resistance patterns against various antibiotics. This information could provide critical insights for clinical decision-making and treatment strategies.
Overall, while the article presents valuable research on this important topic, addressing these points could significantly enhance its impact and utility for both researchers and clinicians in the field of diabetic foot sepsis management.
Author Response
|
Response to Reviewer 1 Comments
|
|||
|
1. Summary |
|
|
|
|
Thank you very much for taking the time to review this manuscript. Please find the detailed responses below and the corresponding revisions/corrections highlighted/in track changes in the re-submitted files.
|
|||
|
3. Point-by-point response to Comments and Suggestions for Authors |
|||
|
Comment 1: The paper demonstrates that microbiological testing and antimicrobial resistance profiling in patients with diabetic foot sepsis are similar across many developed and developing countries. While this finding is valuable, it seems that little attention has been given to analyzing the differences between these settings. A more in-depth comparison of these differences could provide additional insights and enhance the paper's contribution to the field.
|
|||
|
Response 1: Thank you for pointing this out. We agree with this comment. Therefore, we have opted to add a paragraph in the introduction about the difference between high and low income countries. However, it is beyond the scope of our research to compare the two in an in depth manner and a further study can possibly do this better.
Is this fine? And if so can I add a paragraph in the introduction [Explain what change you have made. Mention exactly where in the revised manuscript this change can be found – page number, paragraph, and line.] “[updated text in the manuscript if necessary]”
We inserted a paragraph to the manuscript (highlighted in yellow) Page number: 2 Paragraph: introduction – paragraph 3 Line: 56-76 “In developing countries such as South Africa, the management of diabetes and its complications faces significant challenges. A critical factor is the poor understanding of the disease, which results in thousands of patients "suffering from" rather than "living with" diabetes [13]. This gap in knowledge contributes to inadequate self-care, poor dietary habits, reduced physical activity, and suboptimal health-seeking behaviors [under]. Collectively, these factors foster a demographic profile with distinct susceptibility and resistance patterns compared to developed nations [14]. Compounding these challenges are prevalent comorbidities, such as HIV, which increase diabetes risk due to chronic low-grade inflammation and recurrent co-infections [15]. Furthermore, low socioeconomic status exacerbates these issues through chronic undernutrition, potentially impairing insulin production. Limited financial resources also restrict diet diversity, often resulting in reliance on inexpensive, carbohydrate-dense meals, which promote persistent hyperglycemic states [15]. Additionally, the bacterial distribution and infection profiles in these regions differ significantly to developed regions due to variations in geographic factors, personal health practices, antibiotic use, health-seeking behaviors, and medical management practices [profile]. These disparities influence the microbiota composition and infection susceptibility, further complicating the management of diabetes and its associated complications. The interaction between environmental, socioeconomic, and healthcare-related factors underscores the need for context-specific strategies to address diabetes effectively in these settings [14].”
|
|||
|
Comments 2: There appears to be a lack of clarity regarding the nature of bacterial resistance to antibiotics. Specifically, it is not clearly stated whether the bacteria identified in the study are resistant to a single antibiotic or exhibit multi-drug resistance. This distinction is crucial for understanding the complexity of antimicrobial resistance in this patient population. I recommend addressing this aspect more thoroughly in the paper.
|
|||
|
Comment 3: The paper would benefit from a more detailed analysis of the bacterial species most commonly associated with severe outcomes, such as foot amputation or patient death. It would be particularly valuable to include a comprehensive profile of these bacteria and their resistance patterns against various antibiotics. This information could provide critical insights for clinical decision-making and treatment strategies.
|
|||
|
Response 3: Thank you so much for this comment we truly believe it will help us contribute better to patient outcomes however due to our small sample size we are unable to specifically pin point how each organism implicated contributes to a poor prognosis based on their susceptibility and resistance profiles. However we believe that we can contribute in a way that will be fitting by analysing the data in a way that will compare poly and monomicrobial ulcers and their implication on patient outcomes
We inserted a paragraph to the manuscript (highlighted in blue)
Page number: 8 Paragraph: results - paragraph 6 Line: 234 - 239
Outcomes were analyzed based on the microbial profile of ulcers, monomicrobial vs. polymicrobial. Among patients with MCS with no amputations (8), 75% had polymicrobial ulcers, while 25% had monomicrobial ulcers. Among those who underwent amputations (40) 65% had polymicrobial ulcers, and 35% had monomicrobial ulcers. However, statistical analysis (p=0.524) revealed no significant association between the type of ulcer and the outcome (amputation vs. no amputation). 

Page number: 9 Paragraph: discussion – paragraph 4 Line: 281-286
In our research the microbial profile of ulcers being monomicrobial or polymicrobial did not have a statistically significant impact on the likelihood of amputation. This suggests that other factors may be more influential in determining clinical outcomes in patients with diabetic foot ulcers and may be beyond the scope of this research or better extrapolated from a larger sample size which would give a better idea of organism susceptibility and resistance patterns that could frame the outcomes better.
|
|||
Thank you for taking the time to review our manuscript and for providing your thoughtful feedback. We sincerely appreciate the effort you have taken to analyze our work and offer constructive comments.
We have carefully considered your suggestions and made revisions accordingly. We hope that these changes sufficiently address your concerns and enhance the quality of our manuscript. However, if there are any aspects that require further improvement, we would be more than happy to make additional modifications based on your recommendations.
Thank you once again for your valuable input and guidance.
Kindest regards
Dr Simran
Reviewer 2 Report
Comments and Suggestions for Authors
This is a reasonable study.
But the authors need to highlight what they did in those who have drug resistant infections. What is the outcome in these drug resistant cases.
Author Response
|
1. Summary |
|
|
|
|
Thank you for reviewing this manuscript and providing valuable feedback. Detailed responses to your comments are provided below, and the corresponding revisions or corrections have been highlighted using track changes in the re-submitted files.
|
|||
|
|
|||
|
|
|||
|
|
|||
|
|
|||
|
|
|||
|
|
|||
|
|
|||
|
2. Point-by-point response to Comments and Suggestions for Authors |
|||
|
Comments 1: This is a reasonable study. But the authors need to highlight what they did in those who have drug resistant infections. What is the outcome in these drug resistant cases.
|
|||
|
Response 1: Thank you for your comment and for appreciating the study. We acknowledge the importance of understanding outcomes in patients with drug-resistant infections. In our study, we attempted to highlight the clinical outcomes by categorizing patients based on whether they underwent amputation or managed without it. However, we recognize that drawing robust conclusions on the impact of drug-resistant infections is limited by the small sample size and the relatively restricted distribution of susceptibility and resistance profiles in our cohort. This limitation prevents us from making statistically significant extrapolations regarding the outcomes specifically associated with drug-resistant infections. We agree that future studies with larger sample sizes and more diverse microbial resistance data are needed to comprehensively evaluate the outcomes in drug-resistant cases, and we have noted this as a recommendation for further research in our discussion.
|
|||
|
|
|||
|
|
|||
|
|
|||
|
|
|||
|
|
|||
|
|
|||
|
|
|||
|
|
||
|
|
|
|
|
|
||
|
|
|
|
|
|
|
|
|
|
|
|
|
|
|
|
|
|
|
|
|
|
|
|
|
|
|
|
|
|
||
|
|
||
|
|
||
|
|
||
|
|
||
|
|
||
|
|
||
|
|
||
|
|
||
We have therforee added two paragraphs in out results and discussion to touch on this and hope that it is sufficient.
We inserted a paragraph to the manuscript (highlighted in blue)
Page number: 8
Paragraph: results - paragraph 6
Line: 234 - 239
Outcomes were analyzed based on the microbial profile of ulcers, monomicrobial vs. polymicrobial. Among patients with MCS with no amputations (8), 75% had polymicrobial ulcers, while 25% had monomicrobial ulcers. Among those who underwent amputations (40) 65% had polymicrobial ulcers, and 35% had monomicrobial ulcers. However, statistical analysis (p=0.524) revealed no significant association between the type of ulcer and the outcome (amputation vs. no amputation). 

Page number: 9
Paragraph: discussion – paragraph 4
Line: 281-286
In our research the microbial profile of ulcers being monomicrobial or polymicrobial did not have a statistically significant impact on the likelihood of amputation. This suggests that other factors may be more influential in determining clinical outcomes in patients with diabetic foot ulcers and may be beyond the scope of this research or better extrapolated from a larger sample size which would give a better idea of organism susceptibility and resistance patterns that could frame the outcomes better
Thank you for taking the time to review our manuscript and for providing your thoughtful feedback. We sincerely appreciate the effort you have taken to analyze our work and offer constructive comments.
We have carefully considered your suggestions and made revisions accordingly. We hope that these changes sufficiently address your concerns and enhance the quality of our manuscript. However, if there are any aspects that require further improvement, we would be more than happy to make additional modifications based on your recommendations.
Thank you once again for your valuable input and guidance.
Kindest regards
Dr Simran
Reviewer 3 Report
Comments and Suggestions for Authors
Unfortunately, the manuscript is poorly written, starting from the title and abstract, all the way to the discussion.
First of all, in the title is written "patients with diabetic foot sepsis", while in the entire manuscript the authors do not write about sepsis at all. Moreover, in the introduction, lines 72-74, the authors cite the definition of "diabetic foot sepsis" from reference 2, but it is actually the definition of "diabetic foot syndrome". Because of this, I get the impression that the authors do not understand clearly what they are writing about.
The introduction is unnecessarily too long, the methodology is inadequate, the results are poorly analyzed. It is not clear what the authors wanted to show with this research, and there is no possibility to draw any conclusions from the presented results. The names of bacterial species are not italicized and are sometimes written with a small initial letter. Neither the results of antimicrobial testing nor the applied antimicrobial therapy were clearly and unambiguously presented and interpreted.
In my opinion, the research was inadequately conducted, and the obtained results do not bring anything new, and the manuscript as such is not suitable for publication.
Author Response
|
Response to Reviewer 3 Comments
|
||
|
1. Summary |
|
|
|
Thank you very much for taking the time to review this manuscript. Please find the detailed responses below and the corresponding revisions/corrections highlighted/in track changes in the re-submitted files
|
||
|
2. Point-by-point response to Comments and Suggestions for Authors |
||
|
Comments 1: Unfortunately, the manuscript is poorly written, starting from the title and abstract, all the way to the discussion. First of all, in the title is written "patients with diabetic foot sepsis", while in the entire manuscript the authors do not write about sepsis at all. Moreover, in the introduction, lines 72-74, the authors cite the definition of "diabetic foot sepsis" from reference 2, but it is actually the definition of "diabetic foot syndrome". Because of this, I get the impression that the authors do not understand clearly what they are writing about. The introduction is unnecessarily too long, the methodology is inadequate, the results are poorly analyzed. It is not clear what the authors wanted to show with this research, and there is no possibility to draw any conclusions from the presented results. The names of bacterial species are not italicized and are sometimes written with a small initial letter. Neither the results of antimicrobial testing nor the applied antimicrobial therapy were clearly and unambiguously presented and interpreted. In my opinion, the research was inadequately conducted, and the obtained results do not bring anything new, and the manuscript as such is not suitable for publication.
|
||
|
Response 1: We appreciate the time and effort taken to review our manuscript and provide constructive feedback. We would like to address the concerns raised as follows:
We have made the necessary adjustments to the manuscript to address the concerns raised and to improve the overall clarity and structure of our work. Thank you once again for your valuable feedback, which has helped us strengthen the presentation and focus of our research.
|
||
Thank you for taking the time to review our manuscript and for providing your thoughtful feedback. We sincerely appreciate the effort you have taken to analyze our work and offer constructive comments.
We have carefully considered your suggestions and made revisions accordingly. We hope that these changes sufficiently address your concerns and enhance the quality of our manuscript. However, if there are any aspects that require further improvement, we would be more than happy to make additional modifications based on your recommendations.
Thank you once again for your valuable input and guidance it is really helping me on this journey that I have recently started on in the research sphere as a junior doctor.
Kindest regards
Dr Simran
Round 2
Reviewer 1 Report
Comments and Suggestions for Authors
None
Author Response
Thank you very much for your valuable comments. We sincerely appreciate them.
Reviewer 3 Report
Comments and Suggestions for Authors
In this new, revised version of the manuscript, the authors have improved some of its parts; they have made some of the necessary adjustments to the manuscript but have not addressed all concerns and improve the overall clarity of their work. In my opinion, the changes are not sufficient to make the manuscript suitable for publication in the present form.
1. Definition of Diabetic Foot Sepsis
I do not agree with the authors about the use of the terms "sepsis" and "syndrome" and the same should be changed in the entire manuscript.
Lines 32-35: “Diabetic foot sepsis (DFS) is defined by the World Health Organization as a severe complication of diabetes and is characterised by: foot ulceration associated with neuropathy; varying degrees of ischemia; and infection [2]”
Ref 2: Tuttolomondo, A., Maida, C., Pinto, A. Diabetic foot syndrome: Immune-inflammatory features as possible 453 cardiovascular markers in diabetes. World J Orthop. 2015, 6(1), 62-76. doi. 10.5312/wjo.v6.i1.62: “Diabetic foot syndrome (DFS), as defined by the World Health Organization, is an “ulceration of the foot (distally from the ankle and including the ankle) associated with neuropathy and different grades of ischemia and infection”
2. Introduction
Introduction is more cohesive and provide a clearer context for the study. – I agree with the authors
3. Limitations of Retrospective Design
An explanation of the limitations of the study has been added to the discussion. - I agree with the authors
4. Formatting Revisions
Although the authors claim that they have thoroughly reviewed and revised the formatting of the manuscript, bacterial species names are not italicized (in the manuscript, figure and table).
5. Significance of Findings
I agree, and this should be clarified in the discussion.
6. Lines 362-363: “…this finding suggests that macrolides and clindamycin might also demonstrate promising susceptibility against E. faecalis”
Enterococcus faecalis is intrinsically resistant to clindamycin, so this statement is incorrect.
- Gram negative and Gram positive or gram negative and gram positive?
- The entire manuscript needs to be reviewed, there are missing letters in words, parts of the text are bold, etc.
32
ar mar
Author Response
Definition of Diabetic Foot Sepsis
I do not agree with the authors about the use of the terms "sepsis" and "syndrome" and the same should be changed in the entire manuscript.
Answer: Noted. We fully agree with the reviewer. The bigger picture is diabetic foot syndrome.
Lines 32-35: “Diabetic foot sepsis (DFS) is defined by the World Health Organization as a severe complication of diabetes and is characterised by: foot ulceration associated with neuropathy; varying degrees of ischemia; and infection [2]”
Answer: Noted and agreed. The definition is of diabetic foot syndrome and not diabetic foot sepsis.
Ref 2: Tuttolomondo, A., Maida, C., Pinto, A. Diabetic foot syndrome: Immune-inflammatory features as possible 453 cardiovascular markers in diabetes. World J Orthop. 2015, 6(1), 62-76. doi. 10.5312/wjo.v6.i1.62: “Diabetic foot syndrome (DFS), as defined by the World Health Organization, is an “ulceration of the foot (distally from the ankle and including the ankle) associated with neuropathy and different grades of ischemia and infection”
Answer: Noted. We fully agree with the reviewer. We have made correction and to distinguish diabetic foot syndrome from diabetic foot sepsis.
- Introduction
Introduction is more cohesive and provide a clearer context for the study. – I agree with the authors
Answer: Noted. Thank you very much for the valuable inputs and recommendations.
- Limitations of Retrospective Design
An explanation of the limitations of the study has been added to the discussion. - I agree with the authors
Answer: Noted. Thank you very much for the recommendations.
- Formatting Revisions
Although the authors claim that they have thoroughly reviewed and revised the formatting of the manuscript, bacterial species names are not italicized (in the manuscript, figure and table).
- Significance of Findings
I agree, and this should be clarified in the discussion.
Answer: Noted and agreed. We have included the significance of the findings in the discussion.
- Lines 362-363: “…this finding suggests that macrolides and clindamycin might also demonstrate promising susceptibility againstE. faecalis”
Enterococcus faecalis is intrinsically resistant to clindamycin, so this statement is incorrect.
Answer: Noted. We agree with the reviewer and have modified the statement and deleted reference to sensitivity to clindamycin.
- Gram negative and Gram positive or gram negative and gram positive?
Answer: We sincerely appreciate comments of the reviewer. We revised the manuscript and consistently wrote gram-negative or gram-positive.
2. The entire manuscript needs to be reviewed, there are missing letters in words, parts of the text are bold, etc.
Answer: We agree and appreciate the comments of the reviewer. We have reviewed the manuscript and had to correct a number of grammatical and typing errors. Thank you very much.
32